# Understanding factors influencing utilization of HIV prevention and treatment services among patients and providers in a heterogeneous setting: A qualitative study from South Africa

**Lucy Chimoyi**[1,2]*, **Jeremiah Chikovore**[3], **Eustasius Musenge**[2], **Tonderai Mabuto**[1], **Candice. M. Chetty-Makkan**[4,5], **Reuben Munyai**[1], **Tshegang Nchachi**[1], **Salome Charalambous**[1,2], **Geoffrey Setswe**[1,6]

1 Implementation Research Division, The Aurum Institute, Johannesburg, South Africa, 2 School of Public Health, University of the Witwatersrand, Johannesburg, South Africa, 3 Human and Social Capabilities Research Division, Human Sciences Research Council, Durban, South Africa, 4 Health Economics and Epidemiology Research Office (HE2RO), Johannesburg, South Africa, 5 Faculty of Health Sciences, Department of Internal Medicine, School of Clinical Medicine, University of the Witwatersrand, Johannesburg, South Africa, 6 Department of Health Studies, University of South Africa, Pretoria, South Africa

* LChimoyi@auruminstitute.org

## Abstract

Despite advances made in HIV prevention and treatment interventions in South Africa, barriers to their utilization continue to exist. Understanding perspectives from patients and providers of healthcare can shed light on the necessary strategies to enhance uptake of HIV services. A cross-sectional qualitative study was conducted in July 2020 in Ekurhuleni District. Based on HIV prevalence estimates from a national survey, male condom use coverage and antiretroviral treatment (ART) initiation rates from routinely collected clinical data for 2012, we selected facilities from geographical areas with varying HIV prevalence and uptake of HIV services. In-depth interviews were conducted with adult (≥18 years) patients and healthcare workers in selected primary healthcare facilities. Thematic analysis was performed following a framework built around the social cognitive theory to describe behavioural, personal, and social/environmental factors influencing utilization of HIV services. Behavioural factors facilitating uptake of HIV services included awareness of the protective value of condoms, and the benefits of ART in suppressing viral load and preventing mother-to-child HIV transmission which was evident across geographical areas. Barriers in high prevalence areas included suboptimal condom use, fears of a positive HIV result, and anticipated HIV-related stigma while seeking healthcare services. Across the geographical areas, personal factors included ability to correctly use available services enhanced by knowledge acquired during counselling sessions and community-based health promotion activities. Further, social support from family reinforced engagement in care. Compared to low uptake areas, clinics in high uptake areas used care-facilitators, outreach teams and decanting programs to address the environmental barriers including staff shortages and long queues. Barriers at multiple levels prevent optimal utilization of HIV services, calling for strategies that

**Data Availability Statement:** Excerpts of the transcripts relevant to the study are available within the paper and data has been uploaded as supplementary information.

**Funding:** This PhD work is based on the research supported by the Department of Science and Innovation and the National Research Foundation (NRF) through he South African Centre for Epidemiological Modelling and Analysis (SACEMA) www.sacema.org (LC). Any opinion, finding, and conclusion or recommendation expressed in this material is that of the authors and the NRF does not accept any liability in this regard. The funders had no role in study design, data collection and analysis, decision to publish, or preparation of the manuscript.

**Competing interests:** All authors declare no conflicts of interest.

target and address the different levels and tailored to needs of specific settings. Overall, improved delivery of HIV prevention or treatment interventions can be achieved through strengthening training of healthcare providers in facilities and communities and addressing negative sequelae from utilising services in low uptake areas.

## Introduction

The HIV care cascade indicates sub-optimal uptake and coverage of HIV interventions including HIV testing and adherence to antiretroviral therapy (ART) [1,2]. Factors such as anticipated stigma, non-disclosure of HIV status, HIV medication side effects, and low HIV risk perception have been reported worldwide to prevent uptake of HIV interventions [1,3]. Among women, reduced engagement in care has also been noted due to actual or anticipated violent reaction from male partners [4,5]. Men have poorer health outcomes as they are less likely than women to know their HIV status, access and adhere to HIV treatment and more likely die as a result of AIDS-related illnesses due to underutilization of healthcare services [6–8]. Lack of confidential spaces, stock-outs, long queues and negative patient-provider relationship hinder uptake of these interventions [1,3]. Furthermore, lack of awareness on available interventions in the communities and constrained human resources prevent patients from accessing available interventions [3].

Services including HIV testing services (HTS) and ART initiation for those who test positive are a major response to the HIV epidemic, offering a comprehensive package with promising outcomes [9]. In South Africa, primary healthcare clinics (PHCs), often providers of this package, offer HIV pre-and post-test counselling, testing, linkage to care, prevention of onward transmission by providing condoms, pre-exposure prophylaxis (PrEP), post-exposure prophylaxis (PEP), prevention of mother-to-child transmission (PMTCT) and universal test and treat (UTT) [10]. The South African National Strategic Plan (NSP) on HIV, STIs and TB 2017–2022 advocates for the adoption of these available HIV prevention or treatment services to reduce incident infections by more than 60% by 2022 [11].

In many high HIV prevalence settings in Africa, there is heterogeneity in both HIV prevalence [12–14] and uptake of available HIV services [15]. Non-homogenous distribution of resources implies that the dynamics associated with utilization or delivery of HIV programmes may also differ across geographical areas. Limited insights exist regarding these dynamics as they manifest in different geographical areas. Having a context-specific understanding of these factors is necessary to allow realigning of interventions to enhance uptake and coverage of existing HIV programmes [16].

Using a qualitative research approach and drawing on perspectives from patients and providers, this study sought to generate an in-depth understanding of the factors that influence uptake of HIV interventions in a heterogeneous setting in South Africa. The goal is to inform targeted and contextualised implementation to improve uptake.

## Materials and methods

### Study design and theoretical framework

This cross-sectional qualitative study using in-depth interviews (IDIs) sought to understand and describe the factors that influence patients' and providers' utilization of HIV prevention or treatment services [17,18]. We adapted the social cognitive theory (SCT) framework to

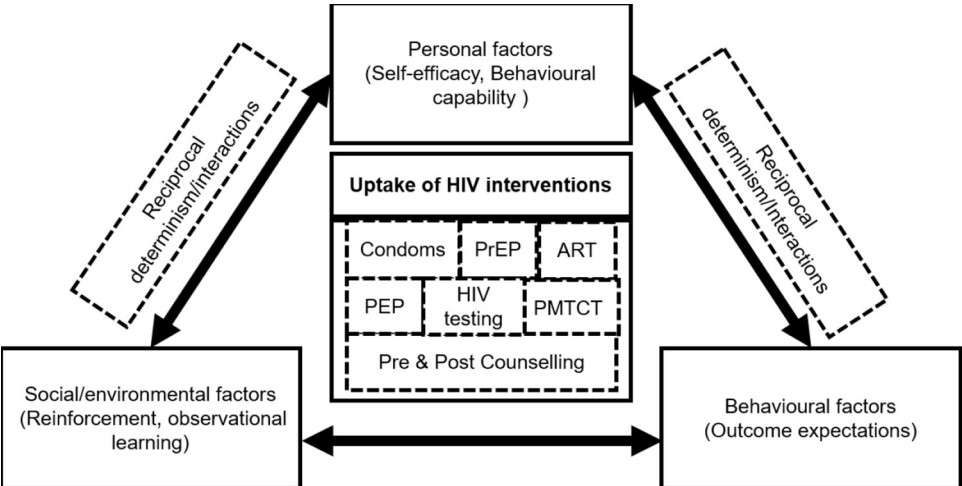

ART: Antiretroviral treatment; PEP: Post-exposure prophylaxis; PMTCT: Prevention of mother to child transmission; PrEP: Pre-exposure prophylaxis

**Fig 1. Adaptation of the social cognitive theory.**

describe patient and healthcare worker (HCW) experiences when utilizing and delivering HIV prevention or treatment services [19]. SCT identifies key determinants of health behaviours including outcome expectations (beliefs about consequences of behavioural choices), behavioural capability (actual ability to perform desired behaviours), self-efficacy (belief in one's ability to account for and check one's behaviours), observational learning (belief based on observing role models accomplish desired behaviours), and reinforcement (responding to external factors). A central concept within SCT is reciprocal determinism, where all factors operate as interacting determinants that influence each other [20]. The adaptation of the SCT theory, used in this study is illustrated in Fig 1 [19].

## Study setting

The study was carried out in three PHCs in Ekurhuleni Metropolitan Municipality (EMM), Gauteng Province, of South Africa. The district is considered high priority in the NSP 2017–2022 due to its large population (3.1 million) [21] and high HIV prevalence (15%) [22]. Primary healthcare clinics serve as the first point of entry in communities where free HIV care and treatment is provided [23]. EMM has a predominantly black male population (51.2%), and almost 25% live below the poverty line (earn less than ZAR 992 per month ≈ US$64) in overcrowded informal settlements on the urban periphery with limited access to job opportunities and adequate social infrastructure [24].

## Sampling, participant recruitment, and data collection

We created three maps showing the distribution of HIV prevalence (estimated from a national survey [25]), and male condom use coverage and ART initiation rates (from routinely collected data in clinics) in 2012. Four geographical areas were identified and categorized as follows: high HIV prevalence and high uptake of interventions (HH); high HIV prevalence and low uptake of interventions (HL); low HIV prevalence and high uptake of interventions (LH); and low HIV prevalence and low uptake of interventions (LL). Participants were enrolled from three clinics in LL, LH and HL areas (Fig 2). LH was further considered as ideal due to the combination of high uptake of interventions and low HIV prevalence whereas LL and HL

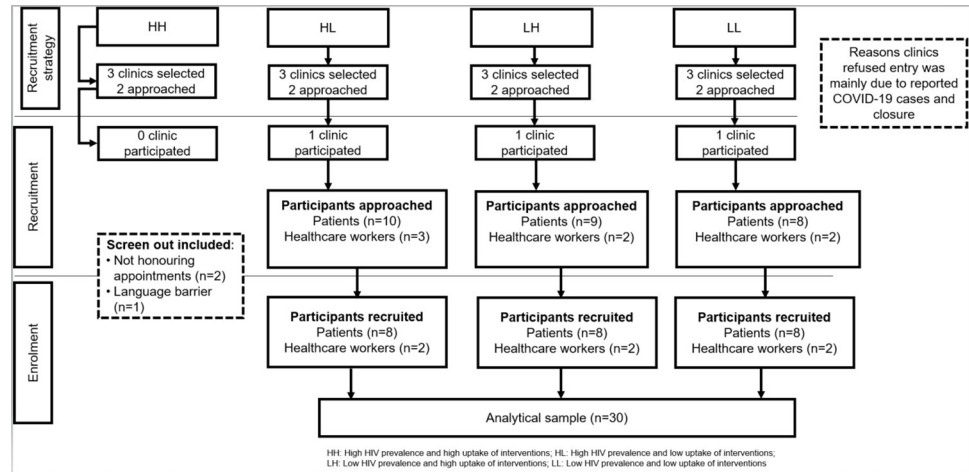

**Fig 2. Flow diagram showing the recruitment strategy and enrolment of participants for the study.**

were considered non ideal areas due to low uptake of interventions. Due to restrictions imposed as a result of COVID-19 pandemic, participants from clinics in HH areas were not included.

Data were collected in July 2020 by four female and three male research assistants (RAs) with experience in conducting qualitative interviews. The RAs, fluent in the major languages used in the study area (English, IsiZulu, Sesotho and Setswana), visited the PHCs during the week between 7am and 12pm. They either approached patients for recruitment as they queued for their routine visits, or as referred by HCWs those newly initiating ART. RAs also made appointments with HCWs providing HIV prevention and treatment services for interviews, which were scheduled at the HCW's convenience. All participants were provided with detailed study information before giving their written informed consent. All participants were either working in or seeking treatment from the selected clinics, ≥18 years and willing to consent to audio recording of the interview. Consenting participants were assigned a unique study number for confidentiality. The final sample of 30 participants included male and female PLHIV, patients who were newly initiated on ART, HIV negative patients, and HCWs providing different HIV services at the clinic and community-level. The variedness of this ultimate sample would enable obtaining a fairly comprehensive picture about experiences and perceptions related to uptake and delivery of HIV services [26].

Interviews lasted 30–45 minutes and were carried out using a guide with open-ended questions. Topics covered in the guide included benefits of using available interventions; knowledge and awareness of HIV prevention and treatment interventions, where to access these and experiences in using them; barriers to and facilitators for uptake, and provider perceptions of regarding the utilization and delivery of HIV care. Interviews were conducted in quiet locations, mostly an open space or in vehicles outside the healthcare facilities, or in empty offices. Discussions were primarily conducted in English, but participants were free to express themselves in vernacular (Setswana, isiZulu or Sesotho) where they felt it helped them better articulate their lived experiences when utilising HIV interventions. RAs were trained to listen carefully and probe during interviews. The investigator (LC) reviewed the first five patient interviews and gave feedback to teams to enhance the questioning and probing. After the interviews, RAs thanked participants and presented a meal voucher of ZAR50≈US$3.3 for their time. Saturation of themes during data collection was achieved through regular debriefing discussions with the RAs on probing techniques [27]. Interviews were stopped when no new issues emerged.

## Data analysis

All interviews were transcribed verbatim. Audio recordings with renderings of local languages were directly transcribed and translated to English by RAs. RM and TN, fluent in the study languages, checked the accuracy of the transcripts against digital recordings. Multiple reading of transcripts was done by LC to capture context, followed by manual coding and categorisation of recurring themes. Transcripts (S1 Data) were imported into QSR International NVivo version 10 software to group the initial codes into themes and subsequently organize into key dimensions and identify patterns across groups. [28]. Soft-copy transcripts were stored securely and safely on password-protected computers and audio recordings deleted from recorders. Transcripts were not returned to participants for comment.

Two members of the study team with Master's and Honours degree qualification (LC and RM) independently reviewed and coded the transcripts guided by the SCT constructs to explore the perceptions of participants on uptake and delivery of HIV services in routine settings. To analyse the qualitative data, we used thematic analysis and inductively and deductively developed codes. The codes were organized into three overarching domains of factors, namely behavioural, personal, and social/environmental. Five themes, aligned to these domains and partly adapted from the original SCT framework and emerging from the data, were defined. SCT constructs were used as initial guides to coding, and the subsequent themes emerging during iterative and deductive coding were therefore closely aligned with the six SCT constructs [29]. Collaboratively, LC and JC (qualitatively -oriented social scientist) reviewed and refined emerging key dimensions and themes The process of refining, reviewing key dimensions and emerging themes was repeatedly done until saturation was achieved when no additional themes or categories could be identified [27]. The analysis process identified salient differences in the geographical settings. Participant demographic characteristics were obtained from the qualitative interviews. We categorized gender based on the responses from the question: "Tell us more about yourself," when the participant explicitly and voluntarily mentioned their gender as either male or female without probing.

## Ethical considerations

University of the Witwatersrand Human Research Ethics Committee (HREC; M181088) granted ethics approval and Ekurhuleni District Research Committee gave permission to access patients and staff in the primary healthcare clinics. All participants provided written consent for participation, audio recording of IDIs and use of their quotations. All participant records and information were anonymized and de-identified prior to analysis.

## Results

### Participants' characteristics

Patients (n = 24) were mostly female (n = 16, 66.7%), not married (n = 16, 66.7%) and HIV-positive (n = 16, 66.7%), and of median age of 37 (IQR: 31–40) years. Most reported having lived in the area for longer than 5 years (n = 15, 62.5%). Slightly above half had been seeking healthcare from the clinics for less than one year (n = 13, 54.2%). HIV-positive patients were on ART for at least 84 (IQR: 21–144) months, and two were newly diagnosed and initiated on ART at the time of the interviews (Table 1). One male patient self-identified as MSM and one HIV-positive female patient was pregnant. The HCWs (all female and employed by either Department of Health (DOH) or donor-funded NGOs working in the area) included a clinician, a professional nurse, a primary health worker, two HIV counsellors and a care facilitator. Most had more than 5 years of professional experience (Table 1). All described their roles as

**Table 1. Socio-demographic characteristics of participants.**

| Patients | N = 24 |
|---|---|
| **Gender** | |
| Female | 16 (66.7) |
| Male | 8 (33.3) |
| **Age**—median (interquartile range) | 37 (31–40) |
| **Self-reported HIV status** | |
| Positive | 16 (66.7) |
| Negative | 8 (33.3) |
| **Duration on ART in months**—median (interquartile range) | 84 (21–144) |
| **Marital status** | |
| Not married | 16 (66.7) |
| Married | 8 (33.3) |
| **Years lived in area of residence** | |
| < 1 year | 1 (4.2) |
| 1–5 years | 8 (33.3) |
| >5 years | 15 (62.5) |
| **Years visiting the current clinic** | |
| < 1 year | 13 (54.2) |
| 1–5 years | 5 (20.8) |
| >5 years | 6 (25.0) |
| **Healthcare workers** | **N = 6** |
| **Professional category** | |
| Clinician | 1 (16.7) |
| Professional nurse | 1 (16.7) |
| Primary health worker | 1 (16.7) |
| HIV counsellor | 2 (33.3) |
| Care facilitator | 1 (16.7) |
| **Professional years of experience** | |
| ≤ 5 years | 2 (33.3) |
| >5 years | 4 (66.7) |
| **Years working in the current clinic** | |
| ≤ 5 years | 3 (50.0) |
| >5 years | 3 (50.0) |

involving HIV care and treatment services including counselling, ART services, health promotion, HIV prevention and HIV awareness and education.

We present five themes from the analysis (Table 2) supported with verbatim, minimally edited quotes. Table 3 provides a summary of the thematic similarities and differences across the three geographical settings

## Benefits from adopting HIV prevention or treatment services

Participants anticipated positive outcomes associated with HIV treatment. The anticipation of improved health outcomes emerged across different geographical areas, particularly, in reference to viral load suppression. Other positive health outcomes included good pregnancy outcomes for HIV-positive mothers, general good health including weight gain for those living with HIV, and remaining HIV-negative or not transmitting HIV to sexual partners.

**Table 2. Themes and key dimensions from in-depth interviews and their relevant SCT constructs and domains.**

| Theme and key dimensions* | Relevant SCT construct and *operational definition*** | Relevant SCT domain** |
|---|---|---|
| **Theme 1: Benefits from adopting HIV prevention or treatment services**<br>*Key dimensions*<br>• Remaining HIV negative by using PrEP or PEP in case of unprotected sexual acts and protected sexual acts by condom use<br>• Suppressed viral load as a result of consistently taking ART<br>• Being in HIV care while pregnant for positive pregnancy outcomes including HIV negative children | **Outcome expectations**<br>*"Expectations of positive health outcomes that are likely from a sustained action when utilising HIV prevention or treatment services"* | Behavioural factors |
| **Theme 2: Potential negative sequelae from utilising HIV prevention or treatment services**<br>*Key dimensions*<br>• Fear of testing HIV positive which may lead to anticipated or enacted stigma<br>• Accessing and using condoms especially for female patients may lead to anticipated or enacted stigma or intimate partner violence<br>• Concerns from providers when patients do not use condoms in favour of PrEP, or not seek healthcare due to distance from healthcare facilities or prolonged waiting times<br>• Side effects from ART may interrupt or stop utilization of HIV treatment | **Reciprocal determinism**<br>*"Interactions between personal and social/environmental factors that causes behaviour change that negatively influences utilization of HIV prevention or treatment services"* | |
| **Theme 3: Well-informed about available HIV prevention or treatment services**<br>*Key dimensions*<br>• Having knowledge on HIV and available interventions, which facilitates utilization of HIV prevention or treatment interventions such as PrEP and PEP.<br>• Using information and skills obtained during pre- and post-test HIV counselling, health promotion activities or campaigns to remain in HIV care<br>• Attitudes towards utilizing HIV prevention or treatment services | **Behavioural capability**<br>*"Having and using the acquired knowledge and skills on HIV prevention or treatment to utilize the available services"* | Personal factors |
| **Theme 4: Ability to correctly use the available HIV prevention or treatment services**<br>*Key dimensions*<br>• To correctly and consistently use condoms for HIV prevention<br>• Taking ARVs to sustain an undetectable viral load and prevent death | **Self-efficacy**<br>*"Having a good understanding of the importance of HIV prevention or treatment services and persistently using them to monitor and control behaviour"* | |
| **Theme 5: Social or environmental dynamics and conditions supporting utilization of HIV prevention or treatment services**<br>*Key dimensions*<br>• Drawing inspiration from peers who have lived positively despite being HIV positive<br>• Social support from family members to continue utilizing the available HIV prevention or treatment interventions<br>• Ensuring supply of condoms in healthcare facilities and in the communities<br>• Structural support by friendly clinic staff who encourage utilization of interventions including to key populations<br>• Differentiated care delivery of interventions for stable ART patients to overcome barriers associated with staff shortages and lengthy queues | **Observational learning**<br>*"Observing from peers and performing desired behaviours from likening their experiences from utilizing HIV prevention or treatment services"*<br>**Reinforcements**<br>*"Encouraging positive changes through interpersonal and structural support"* | Social or Environmental factors |

* Emerged from the data in the study

**Pre-defined and adopted from the SCT framework.

*"By me taking my treatment, I gave birth to a HIV-negative baby. Although I was stressed about it (my HIV status), I learnt that I needed to consult the nurses and doctors at the clinic if I wanted to deliver a HIV-negative baby." (Female, 36 years, HIV positive, LH)*

**Table 3. Overview of similarities and differences in themes across geographical setting.**

| Theme and key dimensions | HL | LH | LL |
|---|---|---|---|
| **Theme 1: Benefits from adopting HIV prevention or treatment services** | | | |
| Being in HIV care while pregnant for positive pregnancy outcomes including HIV negative children | ○ | ○ | ○ |
| Suppressed viral load as a result of consistently taking ART | ○ | ○ | ○ |
| Remaining HIV negative by using PrEP or PEP in case of unprotected sexual acts and protected sexual acts by condom use | ○ | ○ | ○ |
| **Theme 2: Potential negative sequelae from utilising HIV prevention or treatment services** | | | |
| Fear of testing HIV positive which may lead to anticipated or enacted stigma | ○ | | ○ |
| Accessing and using condoms especially for female patients which may lead to anticipated or enacted stigma or violence | ○ | | ○ |
| Concerns from providers when patients do not use condoms in favour of PrEP or when patients fail to seek healthcare due to proximity to healthcare facilities or prolonged waiting times | ○ | | ○ |
| Side effects from ART that may interrupt or stop utilization of HIV treatment | ○ | | ○ |
| **Theme 3: Well-informed about available HIV prevention or treatment services** | | | |
| Having knowledge on HIV and available interventions, which affect utilization of HIV prevention or treatment interventions such as PrEP and PEP. | | ○ | ○ |
| Using information and skills obtained during pre and post-test HIV counselling, health promotion activities or campaigns to remain in HIV care | | ○ | |
| Attitudes towards utilizing HIV prevention or treatment services | | ○ | |
| **Theme 4: Ability to correctly use the available HIV prevention or treatment services** | | | |
| To correctly and consistently use condoms for HIV prevention | | ○ | |
| Taking ARVs to sustain an undetectable viral load/prevent death | ○ | ○ | ○ |
| **Theme 5: Social or environmental dynamics and conditions supporting utilization of HIV prevention or treatment services** | | | |
| Drawing inspiration from peers who have lived positively despite being HIV positive | | ○ | |
| Social support from family members to continue utilizing the available HIV prevention or treatment interventions | ○ | ○ | ○ |
| Ensuring supply of condoms in healthcare facilities and in the communities | ○ | ○ | ○ |
| Structural support by friendly clinic staff who encourage utilization of interventions including to key populations | | ○ | |
| Differentiated care delivery of interventions for stable ART patients to overcome barriers associated with staff shortages and lengthy queues | | ○ | |

○ Denotes similarities across settings.

**Key**: **HL** = High HIV prevalence and Low uptake of interventions; **LH** = Low HIV prevalence and High uptake of interventions; **LL** = Low HIV prevalence and Low uptake of interventions.

*"Since I have been on this treatment, I do not see anything wrong with it. My viral load is undetectable and I see that I am gaining weight." (Female, 32 years, HIV positive, LL)*

*"Before marriage, I always used to check my HIV status every three months to know about myself and to avoid infecting my sexual partners." (Male, 45 years, HIV negative, HL)*

## Potential negative sequelae from utilising HIV prevention or treatment services

This theme was mostly observed in areas where uptake of HIV services was low. Due to anticipated stigma, HCWs revealed that patients avoided disclosure of HIV status by providing incorrect addresses. This led to challenges when tracing and following-up of patients who defaulted on treatment especially in high HIV prevalence and low uptake areas.

> *"Most clients visit the clinic, look at how they are welcomed or how the facility operates. Someone from a different area comes to this clinic to get help, either because of not wanting to disclose their status at their nearest facility to a lot of people that they may know that side. These people come but provide fake addresses which is a bit challenging when you want to trace them after they have defaulted."* (NGO Staff, HIV counsellor, HL)

Further, anticipated and enacted stigma was particularly reported by female patients. This barrier prevented optimum engagement in care. Other barriers included long queues and prolonged waiting times in facilities which impacted patients' livelihoods.

> *"If you are coming to the clinic as a lady and you want to collect condoms for protection, it's very hard because people are looking at you, judging and asking you why you need them. It is better to buy them in a shop but the sales person also thinks that when a lady buys condoms, she is promiscuous."* (Female, unknown age, HIV positive, LL)

> *"The only thing that I have a problem with this clinic is time. You come early around five o`clock in the morning but wait until 12 o`clock to see the nurse. Maybe it is because there are very many people. You finally miss work and some nurses do not provide you with a letter to explain why I missed work."* (Male, 35 years, HIV positive, HL)

IDIs revealed concerns from HCWs in certain areas about delivery of interventions particularly PrEP to patients seeking protection in case of unprotected sexual acts. These concerns are likely to perpetuate stigmatization of patients who choose to use PrEP for HIV prevention.

> *"My opinion is that we are encouraging clients to not use condoms by providing PrEP. People will start engaging in unprotected sexual intercourse with HIV-positive sexual partners and will not use condoms knowing that PrEP is available. This is not good. It's my personal opinion."* (DOH Staff, Professional Nurse, LL)

Lastly, side effects from treatment were mentioned across the geographical areas. Most participants reported experiencing side effects from ART at the beginning of treatment but were encouraged to continue taking their treatment by HCWs.

> *"I started taking my medication and experienced drowsiness. But after three to four months, everything went back to normal as the nurse had said."* (Male, 38 years, HIV positive, HL)

## Well-informed about available HIV prevention or treatment services

Participants across the geographical areas had the necessary knowledge and skills for HIV prevention or treatment from counselling sessions in healthcare facilities, health promotion activities and campaigns in communities. All participants, described condoms as the commonly accessible HIV prevention method either freely from healthcare facilities or pharmacies and shops in the communities. Participants understood the importance of PrEP in HIV prevention

and appeared to see it as having superior levels of safety in protecting against HIV than condoms.

*"PrEP is a pill and condom is a condom. With a condom, there is a lower chance of getting HIV while you are having sex with HIV infected person. Likewise, with PrEP, when you are having sex without a condom you are still protected. Although using condom is much safer, there are chances that it may break and you get infected." (Male, 40 years, HIV negative, LH)*

There seemed to be adequate knowledge of PrEP among study participants. Patients seeking PrEP services from clinics explained in detail the HIV testing and pre-and post-counselling sessions

*"I usually come to this clinic, for PrEP. Before they give you PrEP, you test for HIV. If negative, the nurses give you pills that you will take for a period to prevent you from being infected. It does not actually motivate you to engage in unprotected sex but if you sometimes have unprotected sex, that pill will protect you." (Male, 26 years, HIV negative, LL)*

Knowledge of PEP services and its access was explored. Most participants knew where to access these services in case of an accidental exposure from an infected partner. However, HIV-positive participants seemed to have limited knowledge on PEP.

*"I'm not sure about PEP but probably immediately after finding out that you had intercourse with an infected person, you go the nearest clinic where they give you something. What I don't know is for how many hours, minutes or days." (Female, 25 years, HIV positive, HL)*

There appeared to be lack of adequate knowledge on universal test and treat (UTT) among the participants. Across all geographical areas, HIV-negative participants showed limited understanding of UTT and its benefit to reduce the risk of onward transmission by supressing viral load whereas some HIV-positive participants misunderstood the benefits of UTT in promoting disclosure.

*"The advantages of UTT is for people to know their HIV status but if you are afraid of telling each other, try to tell one person you trust so that he can be the support. To remind you when to take your treatment and when to go collect them." (Female, 35 years, HIV positive, LL)*

Lastly, discussions revealed that utilization of available interventions was said to depend on attitude and HIV risk perception.

*"Using these interventions depends on my attitude towards HIV. As I said, I always use protection. If you are HIV-negative and do not take precautionary measures, you will become infected with HIV." (Female, 40 years, HIV negative, LH)*

### Ability to correctly use the available HIV prevention or treatment services

Aligned to the self-efficacy construct, this theme focused on participant ability to utilize HIV prevention or treatment services, the barriers and the strategies that would be adopted to overcome identified barriers. Consistent use of condoms to protect oneself and partners from infection or re-infection was commonly mentioned.

*"Yes, I use condoms all the time because I don't want to get re-infected. My viral load is undetectable and I would like it to stay like that." (Female, 30 years, HIV positive, LH)*

HIV-positive participants across geographical settings understood that non-adherence to treatment could result in weight loss, severe sickness, or death. Discussions revealed that to avoid experiencing these negative outcomes, adherence to ART was important.

*"ART to me is like a battery, when it's flat you must charge and if you don't charge, it will shut down. I cannot stop taking treatment because I will lose my body, I am going to be sick and maybe after 2 or 3 years, I will be gone." (Male, 40 years, HIV positive, HL)*

To ensure that patients were correctly utilizing the prevention services, HCWs educated them on proper use and storage of condoms to prevent deterioration in effectiveness and quality.

*"We teach patients how to use condoms. There are many cases where condoms have burst and upon investigation, you find, it's because of poor storage, using expired condoms or reusing them. Our people need education on proper use or storage of condoms. They should not put them in the pocket, walk around the mall or keep the packet in the sun." (DOH Staff, Clinician, HL)*

## Social or environmental dynamics and conditions supporting utilization of HIV prevention or treatment services

Participants were motivated to utilize the available HIV treatment services from learning from and observing peers, as explained in this quote.

*"This person was born HIV-positive and I didn't know I will end up like him. I started reading his story while in school and when I discovered that I am HIV-positive, I was motivated to take my medication as we were born the same way." (Male, 40 years, HIV positive, LH)*

Social support was associated with the successful engagement in HIV care. Family members, particularly parents and siblings emerged as a key source of support, for HIV-positive participants. However, due to fear of disclosure, treatment was interrupted when away from home as explained by one young participant.

*"At home, my parents and siblings remind me to take my medication. When I decide to sleep away from home, I don't bring my medication with me and skip my dose because I don't want anyone apart from my family to know my condition." (Female, 25 years, HIV positive, HL)*

Although there were concerns about enacted, anticipated stigma and discrimination, many participants found strength from the support of those with whom they had close and trustworthy relationships. Family and HCWs increased uptake. However, in some instances, intimate partners prevented utilization of certain services such as condoms despite knowledge of HIV status. HCWs reported the challenges female patients faced from male sexual partners regarding condom use.

*"Every day in the morning, we issue out a new box of condoms which patients take them but do not utilize. The married ones say that their husbands are refusing to use condoms. We counsel the couple, they agree to use them but after a week, their story changes and they tell you that they cannot be told how to manage their bedroom issues." (DOH staff, Professional Nurse, LH)*

Despite challenges in using condoms or ART due to relational dynamics or side effects, it emerged that at a structural level, a positive patient-provider relationship and consistent

supply of ART or condoms at clinic-level encouraged utilization of HIV services. Most HIV-positive patients understood that ART was lifelong and facilities have ensured that stock-outs were minimized. In high uptake areas, there was an adequate supply of ART to HIV patients and evidence of structural and social support.

*"In this clinic, I've never experienced shortage of ART. I always get my medication on time and the nurses always remind me a week before my visit date. If I am unavailable, someone I trust will collect on my behalf." (Female, 35 years, HIV positive, LH)*

In high uptake geographical areas there are MSM friendly clinics. There is no fear of stigmatization from HCWs, which encourages uptake of HIV servicess by this key population.

*"I like this clinic because it provides me comfort as a "deputy person" (MSM). I come here to express myself and I like that whenever I talk about confidential things, the staff respect my privacy and I do not hear of our conversation in the community." (Male, 40 years, HIV positive, LH).*

To address issues around small clinic spaces and staff shortage, clinics in high uptake geographical areas have utilized differentiated delivery of care programmes where patients stable on ART and virally suppressed collect medication every two months for six months from a specified location to ensure retention in care.

*"We can decant the patient for six months. The patient's viral load should be less than 50 copies/ml. We refer to the central chronic medicine dispensing and distribution (CCMDD) program where they get their medication delivered in a box every two months." (DOH Staff, Professional Nurse, LH)*

HCWs mentioned additional strengthening of existing system to expand HIV prevention services. Clinics in high uptake geographical areas have used ward-based outreach teams (WBOTS) to conduct HTS in the communities. The WBOTS work together with mobile testing clinics to link patients in the community for HIV testing. The WBOTS also distribute HIV self-testing kits.

*"We have WBOTS that serve in the community and mobile clinics that deliver HTS. They conduct HIV testing while WBOTS distribute HIV test kits to people in the community unable to visit the clinic for HIV services to test at home." (NGO staff, HIV counsellor, LH)*

In addition to mobile clinics and WBOTS, some clinics utilize case-facilitators to provide linkage to HIV prevention services. Case-facilitators held campaigns to create awareness about HIV and the available services such as PrEP and HTS. At the time of the study, these services were interrupted due to restriction measures put in place during the COVID-19 pandemic.

*"Before COVID-19, we were conducting campaigns in the community, giving health talks, and issuing condoms. Because of COVID-19 we are no longer campaigning but when a patient comes to the clinic and asks for a test, we will ask for the reasons for HIV testing. For those testing negative and have had unprotected sex, we offer PrEP for prevention." (NGO staff, Case-facilitator, HL)*

## Discussion

These findings highlight the relevance of using SCT framework for strengthening utilization of HIV prevention and treatment services in high priority districts such as EMM. SCT provides a framework for understanding how perceptions about self-efficacy and outcome expectations, influenced by personal, interpersonal and environmental factors, as well as behaviour capability, ultimately affect engagement in care. The interactions between factors at each one of these levels is particularly important for understanding why patients utilize HIV services in heterogeneous settings. The construct of self-efficacy emphasized on overcoming barriers to utilization of the available interventions. The construct of behavioural capability emphasized the need for individuals seeking HIV care and treatment to have a desired target of being well informed and knowledgeable about the HIV interventions and their availability. Newer HIV prevention or treatment methods such as UTT were not well understood in settings with low uptake of services. The current study showed that newly diagnosed HIV patients had received immediate ART initiation but most did not show adequate understanding of UTT. Health promoters and counsellors should aim to increase factual knowledge of UTT. In high uptake settings, there was high HIV risk perception which seemed to motivate engagement in HIV services. Findings highlight examining personal and relational dynamics which are likely to influence disengagement along the HIV care continuum.

The various outcome expectations associated with utilizing HIV prorammes in this study have been widely reported elsewhere [16,30,31]. Implementers need to continually consider these factors when rolling out programs at facility level. For instance, HIV-positive women of child bearing age are more motivated to give birth to HIV-negative children as observed in the Option B+ study conducted in four sub-Saharan countries [31]. A similar finding was made in our study, and was observed across geographical settings. This points to widespread effective messaging and education which, in turn, likely encourages women to take up and be retained on ART for PMTCT and to protection of their infants [30]. Corroborating this finding, a study in three high HIV burden South African districts found ART initiation rates in pregnant women being higher compared to males and non-pregnant females [16]. HIV-positive men and women were more motivated to remain in care to suppress their viral load for health reasons despite experiencing side effects [30]. Although our study showed that pregnant mothers were adherent in PMTCT programs, a systematic review conducted by Ng'eno et al highlighted common reasons for the poor outcomes experienced by adolescent and young women. This population is often delayed into HIV care due to inadequate knowledge to navigate health system, and stigma from consequences of a positive HIV care result from healthcare workers, family and community [32]. Psychosocial support from healthcare workers and peer support is encouraged to increase engagement in HIV services [33].

Findings also showed that HIV-negative patients were likely to use HIV prevention services including condoms, PrEP and PEP to prevent infection. There was awareness and acceptability of PrEP and willingness to use was mostly reported in males. However, a recent study in South Africa with the youth revealed that there was low awareness of PrEP and males were not likely to use PrEP compared to females due to its daily pill burden [34]. Our study found no concerns about side effects from using PrEP as reported by other populations (MSM) in different settings (Europe) [35,36]. Although reports on the quality of government issued condoms were frequently cited, findings across all geographical settings showed that males mostly utilized condoms to prevent HIV infection. This was encouraging considering the declining proportions of condom use among men reported in the 2020 Global AIDS report [37]. This however was not consistent with findings on younger males in South Africa who have reported performing condomless sex acts for various reasons including impregnating a female partner

as a symbol of prestige and sexual maturity [38]. However, for females, there was evidence of lack of or inconsistent condom use. This finding was corroborated by a systematic review of multiple studies across sub-Saharan Africa that showed that a lower relationship power limits women's decision-making power to mitigate HIV risk [39]. Many females choose relationship maintenance over their sexual health and are reluctant to suggest using condoms to avoid perceptions of infidelity from male partners which may lead to violent reactions [38]. Furthermore, those in long-term relationships see lack of condom use as proof of commitment to male partners, an assertion that corroborates the gendered power dynamics influencing inability by females to negotiate condom use. A meta-analysis showed improved partner communication was likely to increase condom use frequency among sexual partners [40]. In addition, good quality and desirable condoms to the end-user should be distributed in government healthcare facilities.

Peer support encouraged uptake of HIV interventions as shown across all settings in our study. Implementers need to encourage patients to explore potential sources of social support continuously, as they are likely to evolve. Newly diagnosed HIV patients entering care are likely to have varying level of support which may increase their vulnerability to disengaging in care [41]. Expanding programs that address the lack of social support from a psychosocial perspective are likely to improve health outcomes. Individual and family-centred counselling, and peer support while implementing mental health services have improved linkage to and retention in care among PLHIV [42]. Studies in South Africa on healthcare utilization highlight structural factors that continue to challenge uptake of HIV services [43,44]. Factors such as long queues at clinics, shortage of staff, and lack of privacy in clinics have been widely mentioned [45]. In Ekurhuleni, structural support is improving HIV outcomes. Our study identified the use of WBOTs and care-facilitators who are expanding HIV testing, a critical entry for linkage and retention [46]. Linkage to HIV prevention services such as PEP, PrEP and condoms are offered to HIV-negative whereas treatment programs to patients living with HIV. Some clinics in Ekurhuleni provide decanting services to stable ART patients, following the differentiated care model to encouraged retention in care. Most of these structural interventions were reported in high uptake areas.

The fear of HIV-related stigma has been well document in existing literature and is shown in this study as a barrier to engaging in HIV care [47,48]. Some patients in low uptake areas provide incorrect contact details and have disengaged from care due to stigma [49]. Other studies found that due to victimisation from community or fear of violence from partners, women were less likely to use condoms for protection from HIV [4,5]. The social stigma around women who seek condoms as a means of HIV prevention is concerning because the perception of promiscuity is a major barrier to utilize this method of HIV prevention which has been reported in HIV prevention studies in South Africa [50]. Offering alternatives such as PEP or PrEP is a significant facilitator for uptake of HIV prevention. Similar to condom use, misconceptions towards PrEP use needs to be addressed to curb stigma from partners and community. While offering these alternatives, HCWs should also be trained to avoid perpetuation of social stigma associated with choosing to use PrEP over condoms. Such training should encourage HCWs to view PrEP as an aid that allows control over one's sexual health [51]. Incorrect and inadequate knowledge is a significant barrier to the uptake of interventions, has been linked to perceived stigma and discrimination which likely influences attitudes that may deter engagement in HIV care [52]. Stigma and discrimination influences negatively on testing, disclosure, treatment initiation and adherence [1,52,53]. Community-based interventions including integration of HIV services into existing programs can address the negative sequelae and reduce stigma through sensitization by community healthcare or peer workers and is likely to improve uptake of HIV services, thus leading to higher ART coverage and viral load

suppression [54,55]. Most PLHIV protect themselves from the impact of stigma and discrimination by non-disclosure of status or attending clinics far away from where they live which affects retention in care [56]. Strategies that reduce stigma and discrimination are required for continuous engagement in care. Robust education campaigns are needed for both patients and providers to demystify misconceptions about existing HIV prevention strategies and are necessary for a successful demand-creation strategy.

Limitations of the study include the sample comprising of patients already seeking care at selected clinics in Ekurhuleni District. The findings may have limited generalizability to populations that did not attend clinic visits and outside Ekurhuleni. We also acknowledge that the study was conducted in an area where research is frequently conducted, and participants may have increased exposure to HIV information and health services. Nevertheless, we believe participants provided real experiences that validated our findings, as the qualitative data collected was comparable to data provided in other settings examining the same topics, which further attenuated this concern. The study was conducted four months into the COVID-19 lockdown and some clinics did not provide access to participants due to closure after reports of possible exposure or not permitting non-essential activities including research. However, we believe that the findings are a reflection of a heterogeneous setting. Despite these limitations, the strength of this study is that it is one of few studies to explore how SCT constructs amplify the experiences of patients as they engaged in HIV care while looking at geographical differences. The findings are likely to reinforce and expand the existing programmes that encourage utilization of HIV prevention or treatment services and emphasize the relevance of the SCT in the implementation of HIV prevention and treatment programs.

## Conclusions

Firstly, negative sequelae such as HIV-related stigma is a significant problem for implementers of programs because it reduces engagement in care. Decreasing stigma by increasing knowledge and encouraging adequate support network are key to successful HIV prevention or treatment programs in Ekurhuleni. Secondly, mass campaigns and health talks on proper storage, consistent and correct condom use are strongly encouraged to increase self-efficacy. Thirdly, gender-transformative approaches addressing low relationship power in females are useful in encouraging decision-making in improving protective sexual behaviours. Lastly, to address context-specific differences in uptake of HIV care in Ekurhuleni, we should take into account the contextual environment in which these services are to be implemented. High uptake areas showed a comprehensive approach to increasing access and uptake of available services. Interventions targeting health system barriers including confidentiality concerns, stigma and staff shortage seen in low uptake areas can increase uptake and delivery. We acknowledge that there was adequate supply of HIV services in the study area. However, in settings where supplies may be irregular, we highlight the need for the healthcare system to include HIV prevention supplies as part of their essential commodity security plans to expanding access. Furthermore, on-going monitoring and evaluation will be needed to plan for further improvements in implementation of HIV and other health services.

## Supporting information

**S1 Data.**
(ZIP)

## Acknowledgments

We are grateful to the study participants for their participation.

## Author Contributions

**Conceptualization:** Lucy Chimoyi.

**Data curation:** Lucy Chimoyi.

**Formal analysis:** Lucy Chimoyi, Jeremiah Chikovore.

**Funding acquisition:** Lucy Chimoyi, Eustasius Musenge.

**Investigation:** Lucy Chimoyi.

**Methodology:** Lucy Chimoyi, Jeremiah Chikovore, Tonderai Mabuto.

**Project administration:** Reuben Munyai, Tshegang Nchachi.

**Resources:** Reuben Munyai, Tshegang Nchachi, Geoffrey Setswe.

**Supervision:** Eustasius Musenge, Salome Charalambous.

**Validation:** Jeremiah Chikovore, Geoffrey Setswe.

**Writing – original draft:** Lucy Chimoyi.

**Writing – review & editing:** Lucy Chimoyi, Jeremiah Chikovore, Eustasius Musenge, Tonderai Mabuto, Candice. M. Chetty-Makkan, Salome Charalambous, Geoffrey Setswe.

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
