## [Decision Letter · Decision Letter 0]

19 Jul 2021

PGPH-D-21-00055

Understanding factors influencing utilization of HIV prevention and treatment interventions among patients and providers in a heterogeneous setting: a qualitative study from South Africa

Dear Dr. Chimoyi,

Thank you for submitting your manuscript to PLOS Global Public Health. After careful consideration, we feel that it has merit but does not fully meet PLOS Global Public Health’s publication criteria as it currently stands. Therefore, we invite you to submit a revised version of the manuscript that addresses the points raised during the review process.

Editor comments:

I appreciated the opportunity to review this well-written manuscript. I encourage you to address all of the reviewers' comments below, particularly concerning the methods and discussion section. In addition, Reviewer #2's point about how setting/sampling seems to disappear from the paper after the methods is important. You spend some time setting up the sampling approach in the methods, but then do not reference it again.The authors should clarify their use and definition of phenomenology. Typically, phenomenology requires more than one-time in-depth interviews with participants. There is usually much deeper involvement with participants (e.g. multiple interviews, participant observation, participatory methods), as well as more iterative and inductive coding and memo-ing. If this was the case for this study, then please revise the methods to reflect this, and raise this in the discussion. If not, then I would drop the phenomenological label.Although I appreciated the concision of the discussion, I agreed with the reviewers that a little more engagement with the literature would be helpful. In addition to their recommendations, how do your findings relate to the recent large-scale treatment as prevention/UTT/test and treat trials that have been conducted in South Africa and other countries?

We look forward to receiving your revised manuscript.

Kind regards,

Marie A. Brault, PhD

Academic Editor

Journal Requirements:

Additional Editor Comments (if provided):

Reviewers' comments:

Reviewer's Responses to Questions

**Comments to the Author**

1. Does this manuscript meet PLOS Global Public Health’s publication criteria? Is the manuscript technically sound, and do the data support the conclusions? The manuscript must describe methodologically and ethically rigorous research with conclusions that are appropriately drawn based on the data presented.

Reviewer #1: Yes

Reviewer #2: Yes

2. Has the statistical analysis been performed appropriately and rigorously?

Reviewer #1: N/A

Reviewer #2: N/A

3. Have the authors made all data underlying the findings in their manuscript fully available (please refer to the Data Availability Statement at the start of the manuscript PDF file)?

Reviewer #1: Yes

Reviewer #2: No

4. Is the manuscript presented in an intelligible fashion and written in standard English?

Reviewer #1: Yes

Reviewer #2: Yes

5. Review Comments to the Author

Reviewer #1: 1. Abstract, line 2: progress” or advances/innovations?

2. Abstract line 16: “inadequate know-how on effectively using condoms”. I’m not convinced this is the best or appropriate phrasing

3. Page 2, line 31: “proponents” does not seem the best choice of word here – they provide the package, but aren’t necessarily active advocates of it

4. Page 5, line 98: Sentence doesn’t make sense. Revise

5. Page 5, line 100: more detail on study languages would be helpful. Provide some context of the socio-cultural setting, ethno-linguistic characteristics of participants etc

6. Page 6, line 132: explain the term “finer coding”

7. Page 7, line 155: How was gender of participants determined? Were they presented with binary categories to self-select? Were there any other gender categories available? Was this self-identified gender?

8. Page 7, line 155: define the term “single” as a characteristic – how was this defined for participants and how were people characterised?

9. Page 7, line 156: how was HIV status determined? Self-reported?

10. Pahe 7,line 60: how did the participant self-identify as MSM? Did he disclose having sex with other men, or did he self-identify with the label “MSM” ?

11. Discussion: more engagement with the dynamics of condom use and condom decision making might be helpful, particularly when discussing the gendered aspect of condom use and motivations – see suggested literature below.

12. Discussions of condom use also lack acknowledgement of different types of condoms, male/female, government provided, bought condoms etc.

13. What about patient perspectives of PrEP side effects? Fears o around PrEP use, stigma of PrEP being seen as ART etc -engage more with the current literature on PrEP

14. Demonstrate awareness of complexity of HIV risk perception, how this links to HIV cascade – i.e. fear, lack of concern, belief that getting HIV is not an issue since there are ARVs. Complexity of risk perceptions and motivations around HIV prevention.

15. Discussions around social support could benefit from acknowledgment of need for holistic interventions addressing psycho-social aspects related to HIV as well. What about mental health and wellbeing, links to treatment adherence and HIV cascade?

16. Discussion around complexity of HIV related stigma, and efforts to reduce stigma- i.e. resulting in lack of care about getting infected with HIV? HIV stigma in South Africa is not static – it is ever changing and adapting to new HIV prevention and treatment landscape. Authors need to demonstrate an awareness of the fast changing context, and how HIV stigma is situated in this. Framing of “need to reduce stigma” is too simplistic, and needs to be unpacked.

17. Authors could demonstrate a closer engagement with the literature. Some suggested readings below:

- Atujuna, M., Newman, P. A., Wallace, M., Eluhu, M., Rubincam, C., Brown, B., & Bekker, L.-G. (2018). Contexts of vulnerability and the acceptability of new biomedical HIV prevention technologies among key populations in South Africa: A qualitative study. PLoS ONE, 13(2), e0191251–17. http://doi.org/10.1371/journal.pone.0191251

- Duby, Z., Jonas, K. McClinton Appollis, T., Maruping, K., Dietrich, J. & Mathews, C. (2021) “Condoms Are Boring”: Navigating Relationship Dynamics, Gendered Power, and Motivations for Condomless Sex Amongst Adolescents and Young People in South Africa, International Journal of Sexual Health, 33(1). DOI: 10.1080/19317611.2020.1851334

- Duby, Z., Jonas, K. McClinton Appollis, T., Maruping, K., Dietrich, J., Vanleeuw, L. & Mathews, C. (2020) “There is no fear in me … well, that little fear is there”: dualistic views towards HIV testing among South African adolescent girls and young women, African Journal of AIDS Research, 19(3). DOI: 10.2989/16085906.2020.1799232

- Gause, N. K., Brown, J. L., Welge, J., & Northern, N. (2018). Meta-analyses of HIV prevention interventions targeting improved partner communication: effects on partner communication and condom use frequency outcomes. Journal of Behavioral Medicine, 41(4), 423–440. http://doi.org/10.1007/s10865-018-9916-9

- Hartmann, M., McConnell, M., Bekker, L.-G., Celum, C., Bennie, T., Zuma, J., & van der Straten, A. (2018). Motivated Reasoning and HIV Risk? Views on Relationships, Trust, and Risk from Young Women in Cape Town, South Africa, and Implications for Oral PrEP. AIDS and Behavior, 22(11), 3468–3479. http://doi.org/10.1007/s10461-018-2044-2

- Hendrickson, Z. M., Naugle, D. A., Tibbels, N., Dosso, A., Van Lith, L. M., Mallalieu, E. C., et al. (2019). “You Take Medications, You Live Normally”: The Role of Antiretroviral Therapy in Mitigating Men’s Perceived Threats of HIV in Côte d’Ivoire. AIDS and Behavior, 23(9), 2600–2609. http://doi.org/10.1007/s10461-019-02614-5

- Irungu, E.M, Ngure, K., Mugwanya, K.K., Awuor, M., Dollah, A., Ongolly, F., Mugo, N., Bukusi, E., Wamoni, E., Odoyo, J., Morton, J.F., Barnabee, G., Mukui, I., Baeten, J.M. & O'Malley, G. & the Partners Scale-Up Project Team. (2020). “Now that PrEP is reducing the risk of transmission of HIV, why then do you still insist that we use condoms?” the condom quandary among PrEP users and health care providers in Kenya. AIDS Care. http://doi.org/10.1080/09540121.2020.1744507

- Lambert, R. F., Orrell, C., Bangsberg, D. R., & Haberer, J. E. (2018). Factors that Motivated Otherwise Healthy HIV-Positive Young Adults to Access HIV Testing and Treatment in South Africa. AIDS and Behavior, 22(3), 733–741. http://doi.org/10.1007/s10461-017-1704-y

- Legemate, E. M., Hontelez, J. A. C., Looman, C. W. N., & de Vlas, S. J. (2017). Behavioural disinhibition in the general population during the antiretroviral therapy roll-out in Sub-Saharan Africa: systematic review and meta-analysis. Tropical Medicine & International Health, 22(7), 797–806. http://doi.org/10.1111/tmi.12885

- Ng’eno, B., Rogers, B., Mbori-Ngacha, D., Essajee, S., Hrapcak, S., & Modi, S. (2020). Understanding the uptake of prevention of mother-to-child transmission services among adolescent girls in Sub-Saharan Africa: a review of literature. International Journal of Adolescence and Youth, 25(1), 585–598. http://doi.org/10.1080/02673843.2019.1699124

Reviewer #2: This study utilized a qualitative approach to explore patients and provider perspectives and to generate an in-depth understanding of the factors that influence uptake of HIV interventions in a heterogeneous setting in South Africa.

Although this manuscript is well written, it does not offer new insights on the subject of focus.

There are some areas that need clarification as follows:

How was the actual recruitment of participants done? When the researcher approached participants at the que in the HIV clinic, what was their reaction? Were there feelings of stigma? Were there difference between recruiting participants through the health providers versus by directly approaching them as they queued for their routine visits? Why were two strategies used to recruit participants?

What do the authors mean by “final sample of 30 participants was diverse enough”? How diverse was this final sample? What do you mean by diverse? Diversity based on what? Please explain.

How did the authors develop the coding framework?

Were there any difference between the HIV positive and negative participants in terms of a particular perspective as presented in the themes? How about differences in self-esteem and utilization of services? Were there differences in motivations for using HIV services by HIV status or by setting?

The authors mention that the married women find it difficult to use condoms and other HIV services. Did the partners of these women know that they were HIV positive? How about the status of the partners for the married women? Was the status for the men known to the couple?

It seems like the participants interviewed had good access to HIV services and as mentioned in the manuscript had a regular supply of ART. What implications do these findings have beyond the study settings, especially in places where supplies may be irregular?

In addition to working with health providers and those infected with HIV to reduce stigma, there is also stigma experienced within the community. Based on their findings, what do the authors suggest should be done to reduce stigma at the community level?

The differences in setting sampled from did not reflect clearly in the findings and the discussion. It would be useful for the authors to present similarities and differences in accounts by settings so that readers can clearly see how the findings are a reflection of the heterogeneous settings as mentioned in the methods and discussion sections.

6. PLOS authors have the option to publish the peer review history of their article (what does this mean?). If published, this will include your full peer review and any attached files.

**Do you want your identity to be public for this peer review?** For information about this choice, including consent withdrawal, please see our Privacy Policy.

Reviewer #1: No

Reviewer #2: No

---

## [Decision Letter · Decision Letter 1]

28 Oct 2021

PGPH-D-21-00055R1

Understanding factors influencing utilization of HIV prevention and treatment interventions among patients and providers in a heterogeneous setting: a qualitative study from South Africa

Dear Dr. Chimoyi,

Thank you for submitting your manuscript to PLOS Global Public Health. After careful consideration, we feel that it has merit but does not fully meet PLOS Global Public Health’s publication criteria as it currently stands. Therefore, we invite you to submit a revised version of the manuscript that addresses the points raised during the review process.

The authors' have done a good job of addressing the comments raised, and the paper is nearly ready for acceptance.The minor revisions requested primarily focus on clarifying a few points in the methods, and ensuring consistency in terminology. In particular, the authors should be sure to address the questions related to the methods.

We look forward to receiving your revised manuscript.

Kind regards,

Marie A. Brault

Academic Editor

Journal Requirements:

Additional Editor Comments (if provided):

The reviewers and I appreciate the authors' revisions. I agree with the reviewers' suggestions for additional minor edits. These revisions are primarily to further clarify the description of the methods, and ensure consistent terminology throughout.

Reviewers' comments:

Reviewer's Responses to Questions

**Comments to the Author**

1. If the authors have adequately addressed your comments raised in a previous round of review and you feel that this manuscript is now acceptable for publication, you may indicate that here to bypass the “Comments to the Author” section, enter your conflict of interest statement in the “Confidential to Editor” section, and submit your "Accept" recommendation.

Reviewer #1: All comments have been addressed

Reviewer #3: (No Response)

2. Does this manuscript meet PLOS Global Public Health’s publication criteria? Is the manuscript technically sound, and do the data support the conclusions? The manuscript must describe methodologically and ethically rigorous research with conclusions that are appropriately drawn based on the data presented.

Reviewer #1: Yes

Reviewer #3: Partly

3. Has the statistical analysis been performed appropriately and rigorously?

Reviewer #1: N/A

Reviewer #3: N/A

4. Have the authors made all data underlying the findings in their manuscript fully available (please refer to the Data Availability Statement at the start of the manuscript PDF file)?

Reviewer #1: Yes

Reviewer #3: Yes

5. Is the manuscript presented in an intelligible fashion and written in standard English?

Reviewer #1: Yes

Reviewer #3: Yes

6. Review Comments to the Author

Reviewer #1: This manuscript is much improved. The authors have made a great effort at engaging with the literature, and have made some important clarifications around terminology employed and methodology. Additional comments:

P2

-line 8 & 17: Clarify what you mean by “areas”

-line 26: human resources – rephrase to “staff”/ “health care providers”?

P4

-line 81: Gauteng Province, “of South Africa”

-line 85: provide US$ equivalent for international readers

P5, line 104: perhaps just say that data were collected in July 2020?

P6, line 138: surely not “back translated” into English as English was not the original language?

P21, line 442: in addition to being good quality, provided condoms should “desirable” – government issued condoms are often viewed as undesirable as they “smell bad” / too thick etc

P22, Line 485: clarify what “this misinformation” refers to

Re. my previous comment:

- How was gender of participants determined? Was this self-identified gender?

authors responded:

“Gender was self-identified when participants were asked to “tell us more about yourself?” During the interviews, we did not use a demographics questionnaire used to collect this data. We however asked the question “Tell us more about yourself”, in the interview guide and we were able to get the demographic information from the responses provided during discussions". - I am still troubled by this – interviewers categorised participants into genders by their own interpretation of what participant said, or assumption about the person’s gender? Unless ppt clearly stated when asked to “tell about yourself” – that “as a woman” / “as a man”. Methods section should include statement about how gender was categorised, by whom

Re response: “participant self-identified with the label “Deputy-person” which he explained it as being an “MSM” during the interviews.” -Did the participant state that “deputy-person” meant a “gay man” or a man who has sex with other men. Be specific and careful using these categories.

Reviewer #3: Find all in the attachment

7. PLOS authors have the option to publish the peer review history of their article (what does this mean?). If published, this will include your full peer review and any attached files.

**Do you want your identity to be public for this peer review?** For information about this choice, including consent withdrawal, please see our Privacy Policy.

Reviewer #1: No

Reviewer #3: No

---

## [Editor Report · Decision Letter 2]

22 Nov 2021

PGPH-D-21-00055R2

Understanding factors influencing utilization of HIV prevention and treatment interventions among patients and providers in a heterogeneous setting: a qualitative study from South Africa

Dear Dr. Chimoyi,

Thank you for submitting your manuscript to PLOS Global Public Health. After careful consideration, we feel that it has merit but does not fully meet PLOS Global Public Health’s publication criteria as it currently stands. Therefore, we invite you to submit a revised version of the manuscript that addresses the points raised during the review process.

Thank you for addressing the comments from Reviewer 1. I apologize that the attachment with the other reviewer's comments was not available previously. Please address these comments, and upload the corrected documents. Please feel free to reach out with any questions.

We look forward to receiving your revised manuscript.

Kind regards,

Marie A. Brault

Academic Editor
---

## [Editor Report · Decision Letter 3]

29 Nov 2021

Understanding factors influencing utilization of HIV prevention and treatment services among patients and providers in a heterogeneous setting: a qualitative study from South Africa

PGPH-D-21-00055R3

Dear Dr. Chimoyi,

We're pleased to inform you that your manuscript has been judged scientifically suitable for publication and will be formally accepted for publication once it meets all outstanding technical requirements.

Within one week, you'll receive an e-mail detailing the required amendments. When these have been addressed, you'll receive a formal acceptance letter and your manuscript will be scheduled for publication.

An invoice for payment will follow shortly after the formal acceptance. To ensure an efficient process, please log into Editorial Manager at https://www.editorialmanager.com/pgph/ click the 'Update My Information' link at the top of the page, and double check that your user information is up-to-date. If you have any billing related questions, please contact our Author Billing department directly at authorbilling@plos.org.

Kind regards,

Marie A. Brault

Academic Editor

Additional Editor Comments (optional):

I appreciate the authors' responsiveness to the reviewers' comments. I believe this manuscript is now ready for publication.